# Metaheuristics-Based Optimization of a Robust GAPID Adaptive Control Applied to a DC Motor-Driven Rotating Beam with Variable Load

**DOI:** 10.3390/s22166094

**Published:** 2022-08-15

**Authors:** Fábio Galvão Borges, Márcio Guerreiro, Paulo Eduardo Sampaio Monteiro, Frederic Conrad Janzen, Fernanda Cristina Corrêa, Sergio Luiz Stevan, Hugo Valadares Siqueira, Mauricio dos Santos Kaster

**Affiliations:** 1Graduate Program in Electrical Engineering (PPGEE), Federal University of Technology—Paraná (UTFPR), R. Dr. Washington Subtil Chueire, 330, Jardim Carvalho, Ponta Grossa 84017-220, PR, Brazil; 2Graduate Program in Industrial Engineering (PPGEP), Federal University of Technology—Paraná (UTFPR), R. Dr. Washington Subtil Chueire, 330, Jardim Carvalho, Ponta Grossa 84017-220, PR, Brazil; 3Electrical Engineering Department, Federal University of Technology—Paraná (UTFPR), R. Dr. Washington Subtil Chueire, 330, Jardim Carvalho, Ponta Grossa 84017-220, PR, Brazil

**Keywords:** GAPID control, bio-inspired metaheuristics optimization, genetic algorithm, particle swarm optimization

## Abstract

This work aims to analyze two metaheuristics optimization techniques, Genetic Algorithm (GA) and Particle Swarm Optimization (PSO), with six variations each, and compare them regarding their convergence, quality, and dispersion of solutions. The optimization target is the Gaussian Adaptive PID control (GAPID) to find the best parameters to achieve enhanced performance and robustness to load variations related to the traditional PID. The adaptive rule of GAPID is based on a Gaussian function that has as adjustment parameters its concavity and the lower and upper bound of the gains. It is a smooth function with smooth derivatives. As a result, it helps avoid problems related to abrupt increases transition, commonly found in other adaptive methods. Because there is no mathematical methodology to set these parameters, this work used bio-inspired optimization algorithms. The test plant is a DC motor with a beam with a variable load. Results obtained by load and gain sweep tests prove the GAPID presents fast responses with very low overshoot and good robustness to load changes, with minimal variations, which is impossible to achieve when using the linear PID.

## 1. Introduction

Currently, most industrial processes still employ traditional controllers such as Proportional–Integral–Derivative (PID), which are simple, practical, have well-established design techniques, and come pre-packaged in most control devices, such as programmable logical controllers (PLCs) [1]. However, linear controllers have limited performance due to their intrinsic linearity [2,3,4]. Nonlinear controllers, in most cases, can easily overcome such limitations [5]. Adaptive control is a class of nonlinear control that takes the linear controller structure and adapts the control parameters dynamically, following a special rule that defines these parameters upon some operating conditions [6]. Other techniques use adaptive practices to adjust functional parameters, such as Fuzzy Adaptive [7,8] and Neural Adaptive [9,10,11,12] control techniques.

Several works propose adaptive PID-based techniques that enhance the controller performance, taking advantage of the robust structure of the traditional PID. One option employs Fuzzy logic to adapt the PID gains. In [13], the membership functions are described for inputs (error and derivative) and outputs ($K_{p}$, $K_{i}$ and $K_{d}$ gains); as such, these gains can adapt according to the designed membership functions. The advantage of using Fuzzy techniques relies on requiring only basic knowledge of how the adaptive gains are expected to perform in the controller. The disadvantage is that the resulting controller is likely not optimal.

Another option consists of building adaptation functions based on the error signal to adapt the PID gains dynamically. In [14], a nonlinear PID (NL-PID) is compared to a PID tuned by a Genetic Algorithm (GA) and a PID adjusted by the Ziegler–Nichols method. This NL-PID takes a Gaussian function of the error and employs linear functions as the adaptive rules for the gains. Such a procedure demands relatively low computing power, but the method to define the adaptive parameters of the NL-PID are not straightforward, and it is unlikely to lead to an optimal controller.

In [15], a nonlinear adaptive PID with parameters defined by Genetic Algorithm (GA) was described. The adaptive PID rules were determined by 11 parameters, representing a complex problem optimized by the GA. Furthermore, its operation needs to calculate five exponential functions for each control iteration which takes a considerable computational power.

The Gaussian Adaptive PID (GAPID) is a kind of adaptive control firstly proposed by this research group in 2015 [16], where the adaptive rule follows a Gaussian function of the system error. A Gaussian curve allows for transition from low to high gain in a smooth fashion. It demands light computational effort by using only one exponential function in each iteration, considering that more specific functions save lots of computing power in massive calculations.

The GAPID control technique currently has no mathematical design procedure. The problem of parameter tuning belongs to the NP-hard challenges, which cannot be addressed with traditional deterministic methods. An alternative to overcome these problems is to use bio-inspired optimization metaheuristics that allow for finding a better controller and eventually optimum parameters in a reasonable period [1]. Such an optimization task poses a particular difficulty because the problem is multi-modal, where distinct solutions have very similar performance values. Furthermore, several solutions are not robust because the system performs well only with the load defined for optimization and decreases considerably with other load values [17].

Regarding bio-inspired metaheuristics, the inspiration for the development of such methods lies, for example, in the Laws of the Darwinian Evolution (evolutionary algorithms) or the collective intelligent behavior of groups of animals (swarm-inspired methods) [18,19,20]. These algorithms are endowed with an intrinsic capability of finding local or, eventually, the globally optimum values of the coefficients of a system.

In this sense, intelligent algorithms have been successfully employed to solve multi-modal optimization tasks from various real-world problems. For example, in [21], the authors proposed the use of an improved version of the Salp Swarm Algorithm to deal with feature selection problems; variations of the Moth-flame algorithm, Particle Swarm Optimization (PSO), Genetic Algorithm (GA), and Bees Life Algorithm (BLA) were addressed for cyber-physical system applications in real fog computing; a hybrid proposition among Firefly Algorithm and PSO is proposed in [22] for optimal cluster head selection in wireless sensor networks with the goal of increasing the energy efficiency of the system; notwithstanding, nonlinear control is the target of many studies involving bio-inspired optimization, such as [23,24,25,26,27,28].

In [17,29] some applications of GAPID were explored by using bio-inspired metaheuristics optimization (Genetic Algorithm and Particle Swarm Optimization (PSO)) to find optimal parameters for the GAPID. In [30] other metaheuristics (Whale Optimization Algorithm (WOA) and Artificial Bee Colony (ABC) targeted to GAPID were also studied, using PSO as a common comparison choice to verify different metaheuristics algorithm effectiveness. This study demonstrated that WOA achieved a slightly better optimization result, but PSO converges faster and presented the smallest dispersion of the final results.

Considering the assumption mentioned above, this work aims to analyze the performances of a set of optimization algorithms in charge of finding optimal solutions for the special case of the GAPID controller. However, differently from the previous works [1,17,29], which applied optimization algorithms to GAPID for the plant with fixed load, the novelty here is that variable load is now being considered in the optimization. This demanded some modifications in the optimization process to result in a more robust controller; in this case, for each iteration and each individual, several loads within the load range were tested, and the mean of the fitness for each load was taken as the individual fitness in that iteration. In practice, such a procedure is capable of leading to a robust controller that is expected to perform optimally within a predefined load range.

To this end, six variations of the Genetic Algorithm and six variations of the Particle Swarm Optimization were selected and applied to a DC motor-driven rotating beam [31,32,33,34,35]. In the first moment, the different optimization strategies to obtain the optimal parameters of the GAPID were examined, then analyzed and compared regarding their efficiencies. Afterward, their performances in an experimental test plant were verified.

The GAPID tuned by optimization algorithms is a robust control technique canning share parameters with a previously designed PID, which facilitates its adoption in the industry but also offers some limitations. In most cases, it has the same restrictions as the PID, such as the windup problem in the integral component and noise amplification in the derived component, which is mitigated in the GAPID for minor errors. The other issue is that the plant must be stressed by a series of tests to obtain the fitness values for each individual in each iteration. It is impractical to perform such a task in a real-world plant, so an accurate model must be derived and the heavy lifting transferred to a computer simulation environment.

The remainder of the paper is divided as follows: Section 2 presents the subjects under study, which are the GAPID control and the optimization algorithms used to find optimal parameters for the GAPID. Section 3 presents the test plant model. Section 4 discusses the methodology employed for simulations and experimental analysis. Section 5 presents the results obtained from simulation, comparisons, and comments about the optimization algorithms and the experimental results obtained in the physical plant. Finally, Section 6 presents the main conclusions and future perspectives.

## 2. Theoretical Foundation

In this section, the GAPID controller technique and Bio-inspired Optimization algorithms that will be utilized in the work are presented, mainly the Genetic Algorithm (GA) and Particle Swarm Optimization (PSO). The Fitness function is also covered, which is very important because it directs the optimization toward the solution. This information is the key to addressing the variations applied in the Optimization algorithm, stated in Section 4, for posterior performance analysis.

### 2.1. GAPID Controller

It is known that nonlinear controllers are more efficient than linear controllers; the problem is that they are harder to design. This work employs the adaptive PID controller, where the proportional, integral, and derivative gains adapt according to a Gaussian function of the input error ε. The conceptual idea is to transition between two gain levels, $k_{0}$ and $k_{1}$, using a smooth process with smooth derivatives as the adaptation rule.

Smooth functions are desirable because they avoid unexpected behavior caused by discontinuous gain transitions. The proposed function is Gaussian-like, as stated in Equation (Equation 1)
(1)$$\lambda \left(\varepsilon \right)=k_{1}-\left(k_{1}-k_{0}\right)e^{-\delta \varepsilon^{2}}$$
where ε is the input error, $k_{0}$ is the gain when |ε|=0, $k_{1}$ is the gain when |ε|=∞, and δ is the openness degree of the Gaussian function. This function will describe the adaptive PID gains $\lambda_{p}\left(\varepsilon \right)$, $\lambda_{i}\left(\varepsilon \right)$, and $\lambda_{d}\left(\varepsilon \right)$ [1].

As one can see, this function has three parameters: $k_{0}$, $k_{1}$, and δ. So, the design process is to define the correct values for these parameters. Figure 1a,b show the function shapes for $k_{0}>k_{1}$, and $k_{1}>k_{0}$, and Figure 1c shows the adjustable concavity degree.

As no mathematical design methodology has yet been developed to find proper parameter values for the GAPID, search algorithms are a good option. In this sense, this work used meta-heuristics optimization to find an optimal solution. The problem has two approaches: the free parameters and the linked parameter cases.

#### 2.1.1. Free Parameters Case

The three PID gains turned adaptive by Gaussian functions become a nine-parameter design ($k_{0}$, $k_{1}$, and δ of Equation (Equation 1) for each of the three gains functions), representing a complex problem to solve. Furthermore, there is the multi-modal problem, where several solutions may result in a near-best controller.

However, this problem can be reduced to one parameter: it is reasonable to state the derivative gain as zero when ε=0, turning the PID into a PI when the output matches the input reference, helping to avoid the problem that noise provokes in the control signal due to the derivative component of PID controllers. Then, the overall problem becomes an eight-parameter design referred to as the free parameters case.

#### 2.1.2. Linked Parameters Case

Considering that most industrial plants utilize PID controllers, several of which are hardly tuned in over long experimentation, switching to a nonlinear adaptive controller with non-tested parameters in a production plant is not easily accepted. In this sense, the proposal of setting an adaptive controller based on the already-determined PID gains is more plausible. The idea is to derive the Gaussian limits $k_{0}$ and $k_{1}$ as functions of the same parameter *x* in the form $k_{0}=x\cdot K$ and $k_{1}=1/x\cdot K$, where *K* is the previously defined linear gain of the original PID. The derived Gaussian functions become
(2)$$\begin{array}{cc} \lambda_{p}\left(\varepsilon \right)= & x\;K_{p}-\left(x\;K_{p}-\frac{1}{x}K_{p}\right)e^{-\delta_{p}\varepsilon^{2}} \end{array}$$
(3)$$\begin{array}{cc} \lambda_{i}\left(\varepsilon \right)= & y\;K_{i}-\left(y\;K_{i}-\frac{1}{y}K_{i}\right)e^{-\delta_{i}\varepsilon^{2}} \end{array}$$
(4)$$\begin{array}{cc} \lambda_{d}\left(\varepsilon \right)= & z\;K_{d}\;e^{-\delta_{d}\varepsilon^{2}} \end{array}$$
where $K_{p}$, $K_{i}$, and $K_{d}$ are the PID gains. Equations (Equation 2)–(Equation 4) represent a six-parameter problem design (*x*, *y*, *z*, $\delta_{p}$, $\delta_{i}$, $\delta_{d}$), which will be referred as the linked parameters case.

As stated in [29], in most cases, the free parameters case leads to a better solution, usually representing a particular best solution that performs very well, but only with specific design conditions. If conditions change, the performance often decreases significantly. On the other hand, the case of the linked parameters, besides performing a little behind the case of the free parameters, is more tolerant to changes and exhibits better robustness to load variations. It also presents the additional advantage of being tied to the original gains that most controllers have already defined in their plants.

### 2.2. Bio-Inspired Optimization

Where finding an optimal solution may be impossible or impractical, heuristic techniques can speed up the process of finding a good one [36].

A meta-heuristic is defined, in computer science, as a high-level heuristic designed to find, generate, or select a search algorithm that could provide some satisfactory possible solutions [37].

In this work, two methods for optimization were used, the Genetic Algorithm (GA) and the Particle Swarm Optimization (PSO). For each of the methods, a set of six variations were examined.

#### 2.2.1. Genetic Algorithm

The Genetic Algorithm (GA) is the first introduced bio-inspired meta-heuristic to deal with optimization problems [38]. Still, it is an essential evolutionary algorithm applied in many areas, such as clustering, time series forecasting, and control [1,17,39].

The GA was inspired by Darwinian evolution by natural selection [40]. Thus, the method is initialized, creating a population of candidate solutions (named individuals or chromosomes) that are randomly generated. Such agents are characterized by their coordinates on the problem-solving space, which means that an individual is a vector containing the values of the addressed problem parameters [41].

Each agent also receives a fitness, that is, a score representing the response quality leading to the system of interest. Bringing this idea to natural selection, solutions with higher fitness values tend to survive the selection process and maintain their genotype in the next generation. After the age of the population and fitness assignment, the first versions of the GA follow three basic steps: selection, crossover, and mutation.

The selection defines the individuals who pass through the other operations. Generally, all methods choose the best-ranked agents, the roulette wheel, and tournament [42].

In the first step, a roulette is created, in which each slice corresponds to an individual. However, the size of such a slice is proportional to the fitness, so that the best-ranked individuals present a higher probability of being selected. It is possible to choose the same agent more than once.

The tournament procedure is initiated by randomly selecting two individuals. Then, they compete in a way that the highest fitness is chosen. In this case, the selective pressure is lower than the roulette wheel [40].

After that, the crossover is applied between the two individuals of the population (parents) selected in the previous step. Such parents change some values of their genes, generating two new individuals (offspring), which replace the parents in the current population. The procedure is repeated until the population presents the same number as the original population. The most known method is the one-point crossover. The crossover is an exploitation procedure or a way to perform local search [38].

The last step is mutation. Usually, it presents a small probability of occurrence and is applied to all the genes of the entire population. Therefore, some may be changed, perturbing their values according to predefined distribution. Another possibility is to disturb the genes considering a predefined number of them. In this case, the probability of occurrence is 100% for the genes selected. This step allows for a global search to find some unexplored regions of the cost function [41]. After that, a new generation is obtained that passes through the same operations until the stop criterion. Figure 2a presents the steps of the traditional GA approach.

Another variation of the GA is based on using a sub-population formed by the sum of parents and offspring during a generation. In this case, the three steps mentioned above occur differently: crossover, mutation, and selection. This approach performs the creation of offspring after the crossover, but the parents remain, doubling the number of individuals in the population. After, the mutation is applied considering the entire population. Finally, the selection (roulette wheel or tournament, for example) is used to select the final population, considering a competition among parents and offspring. Figure 2b presents the steps of the GA-modified strategy.

#### 2.2.2. Particle Swarm Optimization

Particle Swarm Optimization (PSO) is a swarm-based computing technique developed by Kennedy and Eberhart [43], which was inspired by the social behavior of groups of animals, such as flocks of birds and schools of fishes.

PSO is a simple yet powerful optimization tool. Its main characteristic is simple agents working together, which can arise sufficient collective intelligence to conduct the operation, respecting simple rules [44]. Each particle has a position and velocity in the search space. These two variables respect the following rules in each iteration (Equations (Equation 5) and (Equation 6)).
(5)$$v_{i}^{n+1}=\omega v_{i}^{n}+r_{1}c_{1}\left(p_{best}-x_{i}^{n}\right)+r_{2}c_{2}\left(g_{best}-x_{i}^{n}\right)$$
(6)$$x_{i}^{n+1}=x_{i}^{n}+v_{i}^{n+1}$$
where $x_{i}^{n}$ and $v_{i}^{n}$ are position and speed of the *i*-th particle in the *n*-th iteration, *w* is the inertial weight, $r_{1}$ and $r_{2}$ are random numbers between 0 and 1, $c_{1}$ is the personal acceleration factor, $c_{2}$ is the social acceleration factor, $p_{best}$ is the best personal position achieved so far, and $g_{best}$ is the best global position until the current iteration.

Figure 3 presents the steps of the PSO addressed.

Two main topologies are addressed in this work: (a) Global Topology: each particle considers as global best ($g_{best}$) the best-positioned particle for the sake of the entire population; (b) Ring Topology: the particle considers as global best ($g_{best}$) the best-positioned particle among its nearest topological neighbors, which are defined previously.

#### 2.2.3. Fitness Function

The fitness function (FF) represents another critical choice. If a selected FF is inappropriate for a particular application, the result will tend to the best fitness, which may not perform best. In this sense, some common functions that evaluate the performance of the control system were tested. The options are Integral Absolute Error (IAE), Integral Square Error (ISE), and Integral Time Square Error (ITSE).

The test system is a parameterized second-order system:(7)$$P\left(s\right)=\frac{\omega_{n}^{2}}{s^{2}+2\zeta \omega_{n}s+\omega_{n}^{2}}$$

When subjected to a step input, such a system presents a near-best transient when ζ≈0.7. Then, it is reasonable to select a fitness that raises the higher value around ζ=0.7. Figure 4 shows the plots of fitness against ζ.

The Figure 4 evidence IAE has the peak close to 0.7. It also presents a good balance between settling time and overshoot. ITSE is also a good choice but puts a heavier weight on settling time, allowing for higher overshoot. If this is not the case, IAE is the best candidate for FF. In order to normalize the value of the FF, the following definition is adopted:(8)$$\mathrm{Fit}=\frac{1}{1+\mathrm{IAE}},$$
where Fit approaches one as IAE tends to zero.

## 3. Test Plant Model

The application system comprises an axis with a rotating beam linked to a DC motor [31,32,33,34,35]. The control system must be tuned to perform well for load variation defined by different beams, which influences the total moment of inertia of the axis.

With the beams attached, the system is influenced by the drag force, as stated in Equation (Equation 9), which represents an opposite torque to the rotor movement [45].
(9)$$F_{d}=\frac{1}{2}\;C_{d}\;\rho \;A\;v^{2}$$
where ρ is the air density, *A* is the area of the moving object facing the air, $C_{d}$ is the drag coefficient, and *v* is the linear speed.

The state-space equations of the system are given by Equations (Equation 10) and (Equation 11)
(10)$$\begin{array}{cc} L\frac{di_{a}}{dt} & =-Ri_{a}-k_{e}\omega +11v_{pwm} \end{array}$$
(11)$$\begin{array}{cc} J\frac{d\omega }{dt} & =k_{t}i_{a}-B\omega -\frac{{\left(k_{ms}\omega \right)}^{2}k_{acd}}{10} \end{array}$$
where *L* is the stator’s inductance, *R* is the stator’s resistance, *J* is the rotor’s moment of inertia, *B* is the rotor’s friction coefficient, $k_{e}$ is the current-to-torque conversion constant, $k_{t}$ is the speed-to-induced-voltage conversion constant, $k_{s}$ is the linear-to-rotational factor, $k_{acd}$ is the constant equivalent to $C_{d}\rho A$, $i_{a}$ is the stator’s current (state 1), ω is the rotor’s rotational speed (state 2), and $v_{pwm}$ is the input voltage in form of PWM pulses.

From Equations (Equation 10) and (Equation 11), a Simulink model was built in order to do computational simulations, as presented in Figure 5. The plant parameters are summarized in Table 1.

It is highlighted in Table 1 that the moment of inertia J is not fixed because, in this work, it depends on the beam attached to the motor that acts as a variable load. Thus, the load is composed of the moments of inertia of the axis joint and a range of different beams, as summarized in Table 2. This range is used for simulations and covers the beams available in the physical plant.

## 4. Methodology

The methodology for developing this work is divided into two parts: the first determines the procedures for choosing the simulation parameters, starting with the proposal of 6 different scenarios for the GA; and then the six scenarios for the PSO. The best scenario found was used for an experimental evaluation, described in the second part of the methodology.

Thus, a computational model using Matlab/Simulink was elaborated for running simulations. This model comprises the plant and the GAPID controller.

This model is used during the optimization process, where all the algorithms’ variations are executed upon this model, and the desired results are extracted at the end of each simulation.

For each algorithm, each individual (or particle) needs to run the starting transient of the model for each iteration. This means that the model was executed once for *n* individuals, with *k* iterations, considering *h* load variations, for all of the 12 algorithms.

The novelty here is that the load variations represent an additional variable to the optimization problem and then allow for finding an optimal and robust GAPID control. This is important because a large class of plants has variations in their operating conditions that must be considered to achieve a robust stabilization.

The final fitness results, as well as the individual evolution, were collected, tabulated and analyzed in order to verify which algorithm is more appropriate for the problem of finding the optimal parameters of the GAPID.

Some of the best results were tested in the physical plant, the DC motor-driven rotation beam, and one of them, the best one, is described in Section 5.

It is important to remark that the quantity of different loads utilized in the simulations is larger than that used in the physical plant, which allows for a greater definition of the load variation range during the optimization process, which occurs exclusively in the simulation environment.

### 4.1. Methodology for Simulation Analysis

#### 4.1.1. GA Variations

In this work, 6 variations of the GA are used, all of them based on the premises described as follows [38,41,46]:GA1: The first GA follows the classic version initially proposed, except considering the mutation, which always occurs at a low rate. The roulette wheel is used to select the individuals that will participate in the crossover. The same individual may be selected more than once. The one-point crossover occurs with 70% probability, and 5% of the genes considering the entire population are randomly chosen and pass thought the mutation. In this case, it is addressed as a perturbation considering a Gaussian distribution;GA2: The second GA is almost identical, but the crossover probability is 100%. It means that all selected individuals will generate offspring;GA3: This version uses the binary tournament instead of a roulette wheel. However, if an agent loses the game, it returns to the previous population and can be selected again;GA4: This version differs from the last because the “death tournament” is used. It means that if an agent loses the tournament, it is suppressed from the population;GA5: This proposal is more different from the others. The order of the operations is changed so that the crossover happens first, then the mutation, and finally the selection. With this idea, note that an intermediate subpopulation is created with the double individuals (see Section 2.2.1: Ultimately, the selection procedure (roulette wheel) is responsible for choosing the remaining agents, leaving the population with the original size. In this approach, all parents are chosen once at random to perform crossover;GA6: The last GA is similar to the previous one, but a tournament replaces the roulette wheel as a selection procedure.

#### 4.1.2. PSO Variations

Two main topologies are proposed: in the Global Topology, each particle has its social factor $c_{2}$ related to the global best ($g_{best}$), while in the Ring Topology, the particle has its social factor related to its nearest neighbors [41].

Six variations of the PSO are addressed in this study:PSO1: The original proposal from Kennedy and Eberhart [43]. It uses Equations (Equation 5) and (Equation 6) considering the absent of ω, or ω=1. The global topology is addressed.PSO2: It also uses the global topology but includes the inertial weight as a constant value between 0 and 1 [47]. Large values for the inertia are effective for global searching, while small values are better for local searching.PSO3: Another global topology, with a variable inertial weight, decreasing linearly from an initial value to a final one during the iterations, according to Equation (Equation 12), where
(12)$$\omega =\omega_{max}-n\times \frac{\omega_{max}-\omega_{min}}{n_{max}}$$
where $\omega_{max}$ is the initial weight, $\omega_{min}$ is the final weight, *n* is the iteration index and $n_{max}$ is the total iterations. This strategy provides a more efficient way to achieve a faster convergence in the last iterations due to gradually limiting the movement of the particles [48].PSO4: The same as PSO1, but using the ring topology [41].PSO5: The same as PSO2 with the ring topology.PSO6: The same as PSO 3 with the ring topology.

The summary of the examined strategies is presented in Figure 6.

After the simulation scan tests of the scenarios addressed above, the best result will be selected to evaluate it experimentally.

### 4.2. Methodology for Experimental Analysis

To experimentally evaluate the proposed solution, an experimental plant based on a DC Motor with variable load was developed.

The application system comprises an axis with a rotating beam linked to a DC motor. The control system must be tuned to perform well in a broad range of operating conditions. Figure 7 depicts the simplified schematic of the system.

The experimental plant was built with a DC Motor powered by a controlled voltage source passing through the Pololu Dual VNH5019 motor driver shield module. This module is a compact breakout board for ST’s high-power motor driver IC, a fully integrated H-bridge to control the speed of a single brushed DC motor [49]. The motor is connected to an encoder to measure its rotation angle and speed. Figure 8a presents the image of both the motor drive module and the signal acquisition board, and Figure 8b shows the picture of the motor with the encoder attached.

The beams are interchangeable to simulate different loads. Three test beams were used: light slim aluminum, carbon steel, and thick heavy aluminum, as shown in Figure 9. The moments of inertia for these beams are presented in Table 3.

Data from the encoder are sent to a PLC CompactRIO^®^, a real-time embedded industrial Data Acquisition and controller made by National Instruments for industrial control systems. This system consists of a controller with a microprocessor and an FPGA programmable by the user. In our case, the encoder (sensor) signal is connected as a digital input and the PWM signal to control the motor speed as an analog output [50]. Figure 10 illustrates the PLC used.

## 5. Results

### 5.1. Simulation Results

As stated in Section 2.2, six GAs and six PSOs were tested to study the behavior of each algorithm. Each algorithm was run 10 times with the average results taken to compose the figures and tables reported in the following sections. The objective is to observe the characteristics, performance, advantages, and drawbacks of each strategy. Simulations were carried out in Matlab/Simulink employing GA and PSO optimization algorithms. The first step is to compare GA and PSO strategies separately. This action is needed to reach the behavior of selecting different parameters within the same algorithm. In the second step, a general comparison between these two algorithms is made, particularly for the final result, remembering that due to their nature being so different, no fair comparison can be realized regarding the evolution of each algorithm.

### 5.2. Comparison of GA Strategies

As stated in Section 2.2, six GAs were tested to study the behavior of each algorithm. The objective is to observe the characteristics, performance, advantages, and drawbacks. The parameters of each GA employed in the tests are summarized in Table 4.

Figure 11 presents the evolution of each GA regarding the best, the worst, and the mean of all individuals in the population.

To better analyze the overall population distribution at the end of iterations, a boxplot of each GA variant was done, shown in Figure 12.

The final results with the best fitness values are summarized in Table 5.

From Figure 12 and Table 5, some conclusions can be taken: GA3, GA4, and GA6 presented less-dispersed populations in higher fitness ranges, with a smaller dispersion in GA4. This last finding in GA4 may pose a problem because some dispersion is often expected in GAs. The main similarity of these strategies is the use of binary tournament selection. On the other hand, GA1 and GA5, which use the roulette wheel, showed worse results with higher dispersion.

It seems clear that the use of roulette wheel should be avoided to this problem. The literature is abundant in presenting the drawbacks of applying such a selection procedure because it presents a high selective pressure. It means that the individuals with the highest fitness present a high probability of being selected at each iteration. However, this procedure may prematurely guide the search for local minimum points, often far from the global optimum. In this sense, the tournament is a better candidate.

In addition, the use of the death tournament did not perform better than the traditional binary tournament due to the risk of losing potentially well-positioned agents. As seen in Figure 12, despite the dispersion being small in GA4, the top fitness is not the highest.

Regarding creating a sub-population, if the strategy is used with a binary tournament, the results revealed that the algorithm could reach better values to the parameters of the GAPID due to the existence of more individuals simultaneously. This procedure may increase the probability of finding good positions in the search space when applying the genetic operators.

Despite presenting the highest dispersion, GA2 achieved a result comparable to GA3, GA4, and GA6 in terms of the mean and the best individuals. Finally, it is important to mention that the results lie within the topmost 1% of the normalized fitness values. This means that all GAs perform quite well, with slightly different results.

### 5.3. Comparison of PSO Strategies

In addition to the analysis of GA algorithms, six PSO variations were evaluated. The parameters were determined by previous tests and are detailed in Table 6.

As mentioned in Section 4.1.2, PSO 1, 2, and 3 use the global topology while PSO 4, 5, and 6 use the ring topology; variants 1 and 4 use no inertial weight ω while 2 and 5 use fixed ω and 3 and 6 use the linear decreasing weight approach [41].

Table 7 presents the best fitness obtained after 10 simulations for each PSO strategy. It can be noted that the first-ranked strategies employ the global topology, which reinforces the conclusion that this is the best topology. Indeed, for this specific application, the full communication between the particles and the best-positioned one is an advantage, leading to better configurations. It seems clear that the ring topology now allowed the search for the maximum fitness points.

Considering the best combination of strategies used to set ω, linear decay proved to be the best choice, which is an important finding since the modification in the value of ω along the iterations adjusts the model to perform a kind of global search strongly at the beginning of the process (exploration) and to perform a local search refining the solutions along the process (exploitation).

Interestingly, while the global topology led the swarm to small dispersion using all chosen values of ω, its application in the ring topology degraded the performance and increased the dispersion. This is an unexpected behavior since the algorithm tends to present a problematic convergence. This finding reinforces the need to choose the best configuration, considering the specificities of the topology selected.

The Friedman test was applied to the results and resulted in a *p*-value of 0.0027, which means that an optimizer change leads to different results.

Considering the parameters presented in Table 5 and Table 7, simulations were carried out, and all the results were summarized in a boxplot, presented in Figure 13.

This fact evidences the dispersion of results and allows for some statistical analysis that help to assess the differences among the PSO versions.

Due to the nature of the chosen normalized fitness function in Equation (Equation 8), it is important to remark that small values of the cost function IAE result in fitness values close to 1. So, even if the values presented in the graph seem comparable, the performance differences are considerable.

At a first glance, it is clear that the global topology is better for this problem. The performances in these approaches are better than with their ring topology counterparts. The dispersion in these variants is also narrower, which means these algorithms achieve better solutions due to their exploration capability [41].

Comparing the best three proposals demonstrates that using an inertial weight with linear decrease seems to be an advantage.

Another observation is that the topology change results in special performances related to updating velocities. In the most straightforward velocity update strategy, the topology of the ring generates the best results.

Finally, PSOs 5 and 6 are the worst regarding fitness and dispersion.

Figure 14 shows the evolution of each PSO strategy for each iteration. It gives an idea of how the individuals behave during the search. The graphics show the Global Best, the best in each iteration (Iteration Best), the average best of the iteration (Average), and the worse of the iteration (Iteration Worse).

In most cases, 30 iterations are enough for convergence. By the evolution of the average best, PSO2 and PSO3 achieved faster Average and Global Best convergences, indicating fewer iterations would be necessary for the optimization.

The resulting best parameters found, in this case by PSO3, are summarized in Table 8. These parameters were selected and employed to obtain the waveforms shown in Figure 15 and the experimental tests presented in the next section.

It is known that PID has limited robust responses for parameter variations, primarily due to its linear nature, which those nonlinear controllers can overcome. Then, a load sweep was performed to analyze each behavior of the controller, PID, and GAPID, remembering that the GAPID parameters are linked to the PID gains.

In Figure 15, the output transients of the PID and GAPID controlled system subjected to variation of the moment of inertia in the range from 0.002 to 0.015 are presented. As expected, the load variation causes a significant change in behavior in the linear PID. In the GAPID, on the other hand, this behavior change is minimal, providing good robustness to load variation. In this case, there was also a considerable performance improvement. It is important to remark that, in our case, the fitness function is based on IAE, and load variation is considered, showing the robustness behavior becomes the primary objective of the optimization.

### 5.4. Experimental Results

For the experimental analysis, the three beams presented in Section 4.2 were used as loads to the presented physical plant. In it, the motor carries different beams which represent different loads to test the performance and robustness of the controller.

The PID and the GAPID-tuned controllers were tested to a set point of angular speed of 5 rad/s., with the startup waveforms shown in Figure 16 and Figure 17.

For the PID controller, the gain values used were 0.2578 for the proportional ($k_{p}$), and 0.0625 for the integral ($k_{i}$), and the output response for each of the three beams is presented in Figure 16.

In it, one can see that the output speed kept an overshoot under 5% and a settling time under 2 s with the light aluminum beam it was projected for. In contrast, the two other beams could not maintain a low overshoot and took longer to stabilize. The system was not unstable, and the PID was still able to control, but not with the same performance.

For the parameters of GAPID, the best solution (PSO3) parameters were taken, representing the set of parameters shown in Table 8.

In Figure 17, although the slower settling time is close to 4 s, the overshoot was under 5% for the three beams. It is noticeable that the output response behaves similarly even with the load mass variation in comparison to the performance of the PID.

It can be observed that the startup transients shown in Figure 16 demonstrate that only the designed PID for one specific load performs well, while the others exhibit a lack of robustness. Again, the robustness advantage of GAPID is evident, as shown in Figure 17, where the controller performs well for all loads.

## 6. Conclusions

This work addressed a comparative study about two bio-inspired algorithms, Genetic Algorithm (GA) and Particle Swarm Optimization (PSO), with six strategy variations each. These algorithms were used to find the optimal parameters of the Gaussian Adaptive PID control (GAPID) controller.

The test plant is a controlled rotating beam driven by a DC motor. The mathematical model was derived, enabling the building of a computational model used in simulations, and an experimentation plant was used to validate the results.

The GAPID, used to control this plant, represents an NP-Hard and multi-modal optimization problem with several distinct suboptimal solutions, then metaheuristics are used to find optimal solutions to the controller parameters.

In the PSO, the strategies were divided by topology (global or ring topology) and inertial weight factor (absent, fixed, or linearly decreasing).

The quality of the solution was evaluated by a fitness function that employs the Integral Absolute Error (IAE) value. This index covers the settling time and the overshoot of the output waveform.

The computational results were presented in boxplots which allow the analysis of the quality and dispersion of the solutions. It demonstrated that the global topology performs better than the ring topology, with higher fitness values and lower distribution.

A step response test shows how each GAPID optimization performs and how much better they are concerning PID. It is an essential conclusion since the existence of adaptive, robust, and reliable control strategies are very important in real applications.

In future works, it is planned to study a multi-objective optimization approach, which may also help us to better select robust solutions among the near best results. Moreover, other metaheuristic strategies can be tested to determine if any of them are more suitable for this problem, such as Whale Optimization Algorithm (WOA), Artificial Bee Colony (ABC), Differential Evolution (DE), Firefly Algorithm (FA), Bat Algorithm (BA), among others, should be tested in future works. Finally, other plants are also planned to be used as test plants, including higher-order and nonlinear ones.

## Figures and Tables

**Figure 1 sensors-22-06094-f001:**
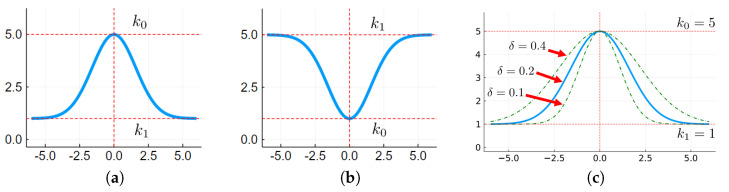
(**a**) Upwards; (**b**) downwards Gaussian function shapes; (**c**) Adjustable concavity.

**Figure 2 sensors-22-06094-f002:**
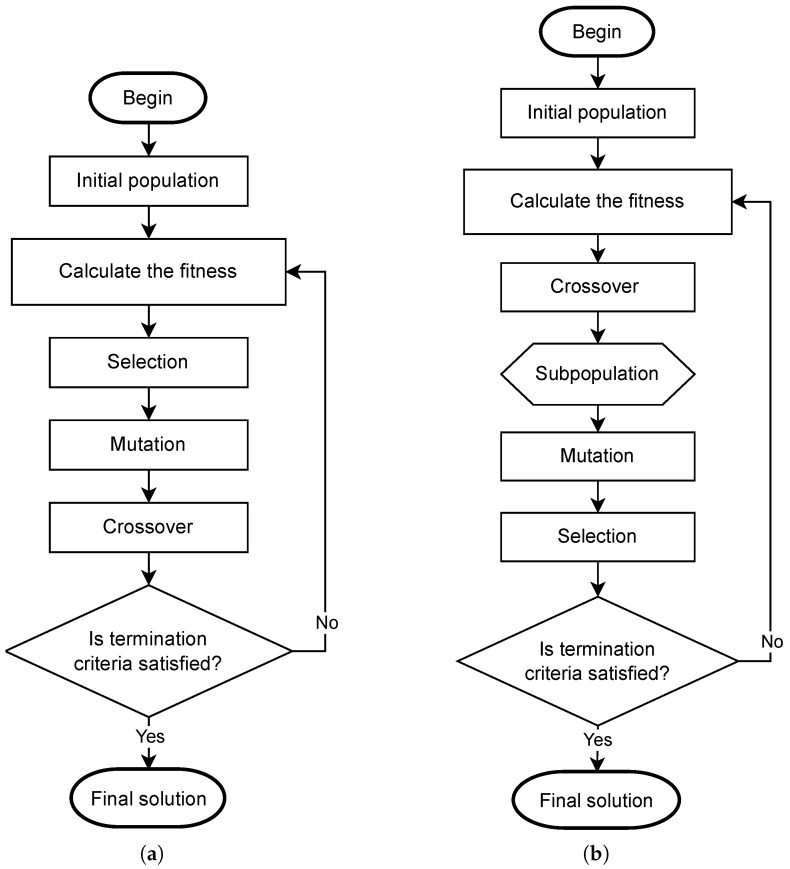
Flowcharts of GA strategies: (**a**) traditional strategy; (**b**) modified strategy.

**Figure 3 sensors-22-06094-f003:**
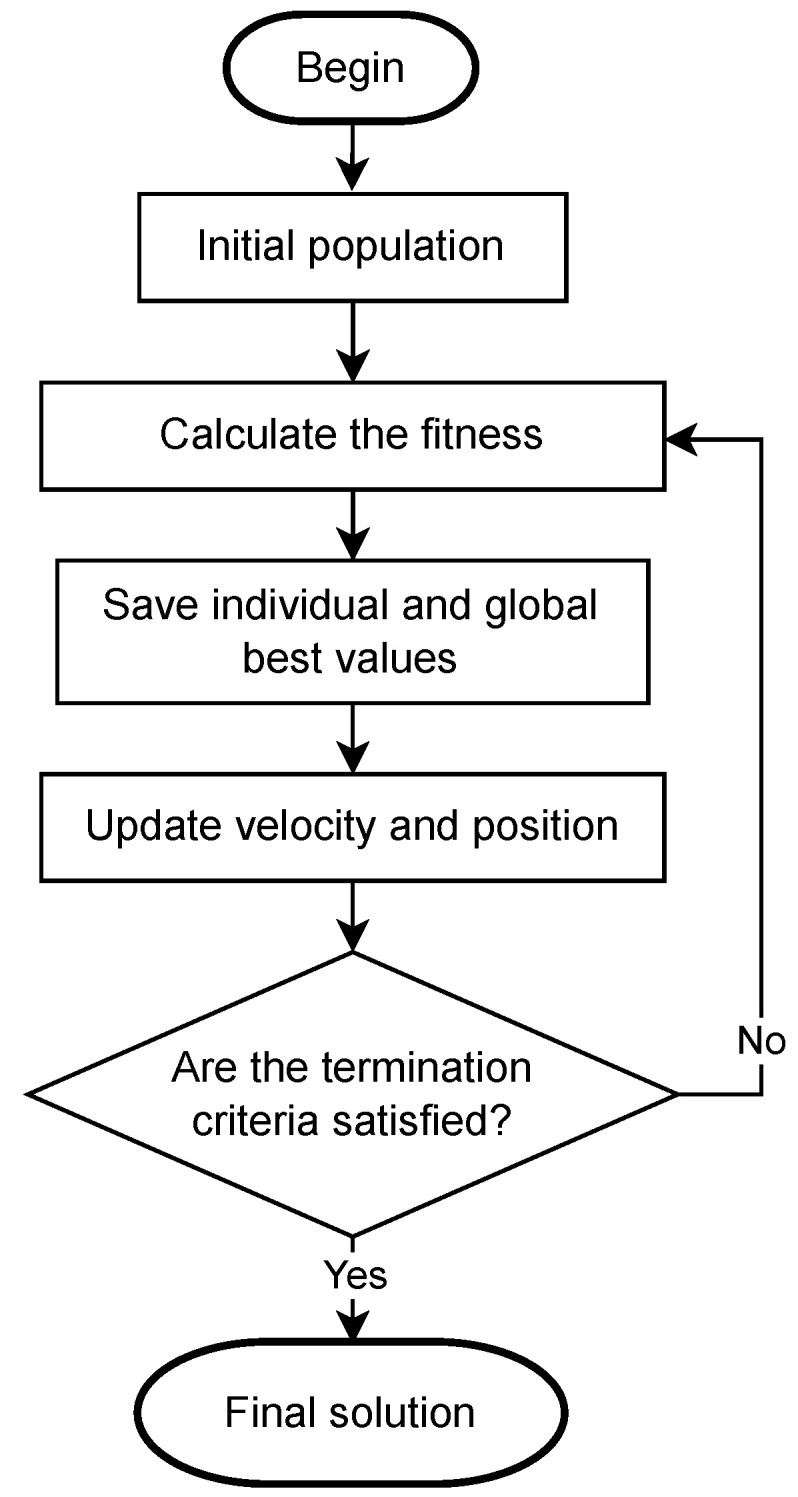
Flowchart of PSO.

**Figure 4 sensors-22-06094-f004:**
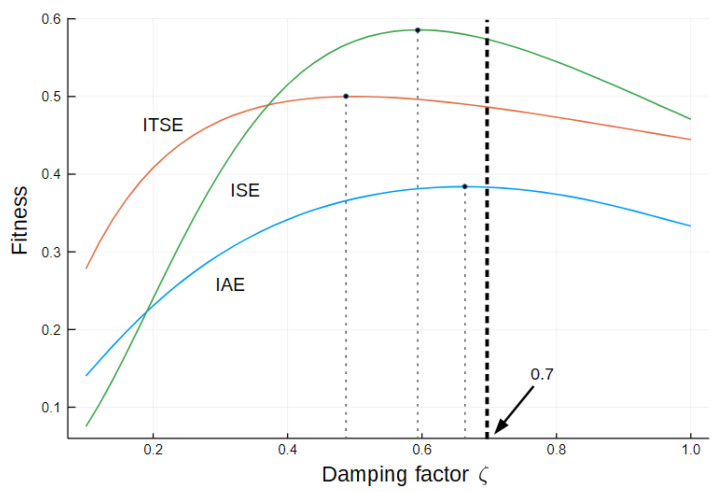
Fitness behavior of IAE, ISE, and ITSE as function of ζ.

**Figure 5 sensors-22-06094-f005:**
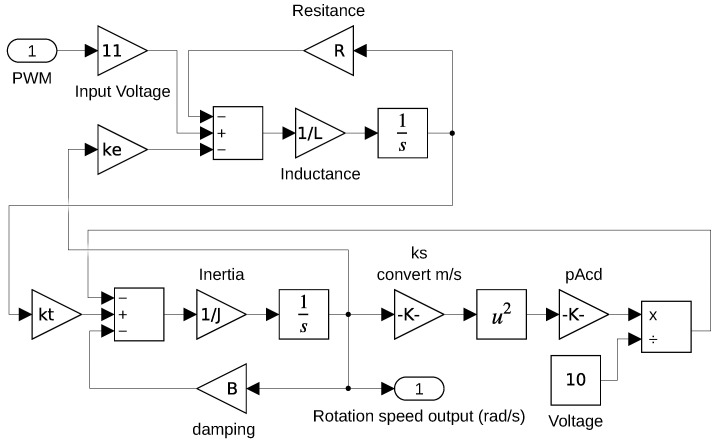
Simulink model of the motor system.

**Figure 6 sensors-22-06094-f006:**
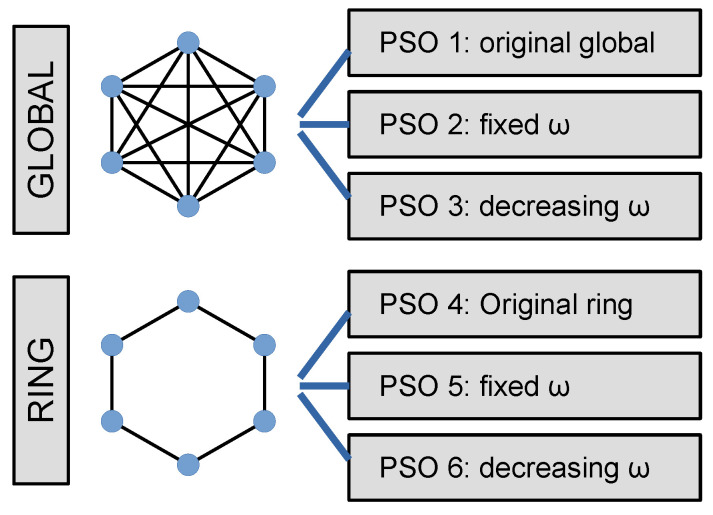
Six PSO strategies.

**Figure 7 sensors-22-06094-f007:**
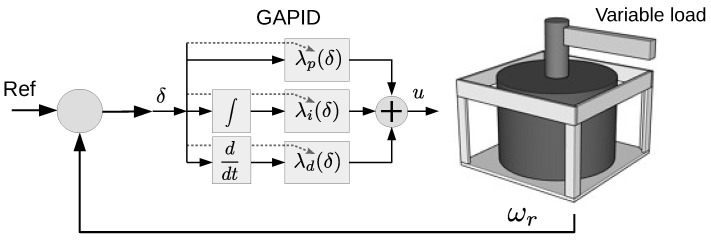
Schematic of the control system.

**Figure 8 sensors-22-06094-f008:**
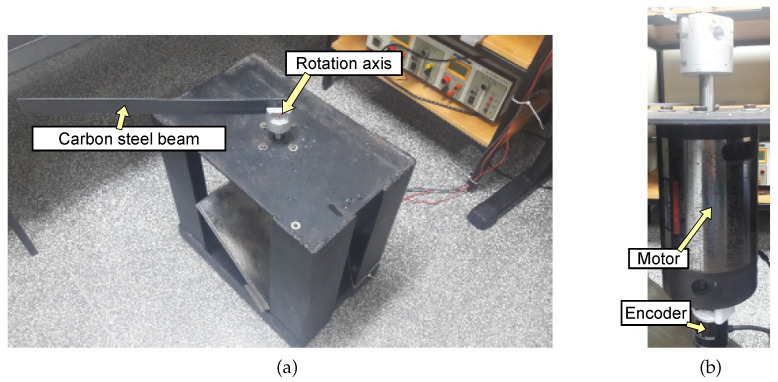
(**a**) Motor system with beam; (**b**) motor with encoder.

**Figure 9 sensors-22-06094-f009:**
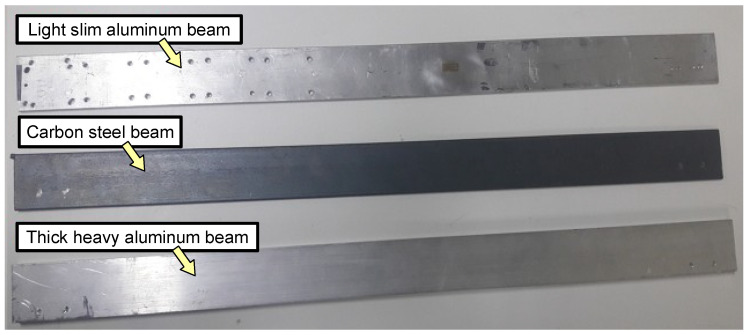
Test beams: light slim aluminum, carbon steel, and thick heavy aluminum.

**Figure 10 sensors-22-06094-f010:**
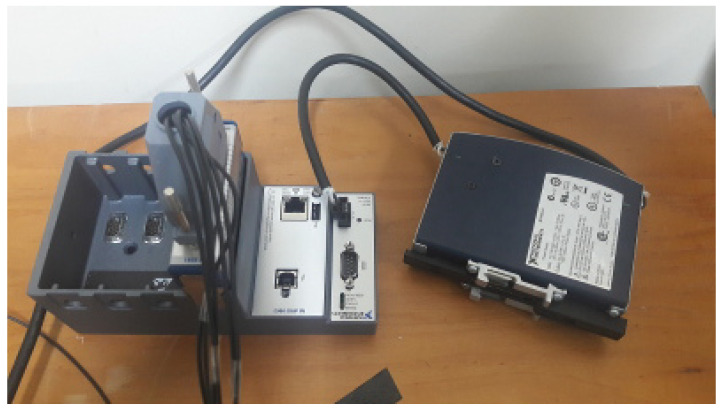
DC motor driver hardware.

**Figure 11 sensors-22-06094-f011:**
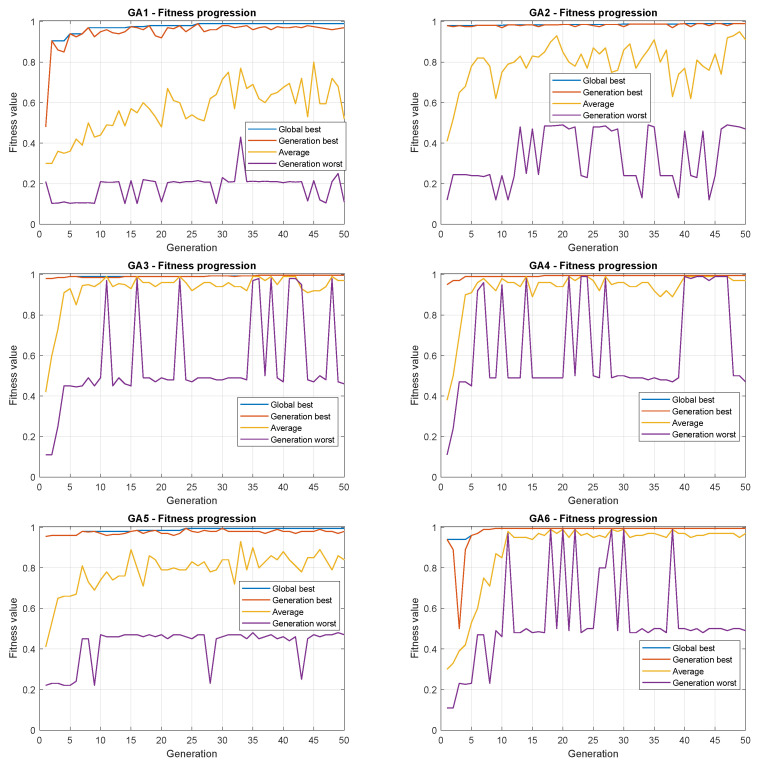
Evolution of GA strategies.

**Figure 12 sensors-22-06094-f012:**
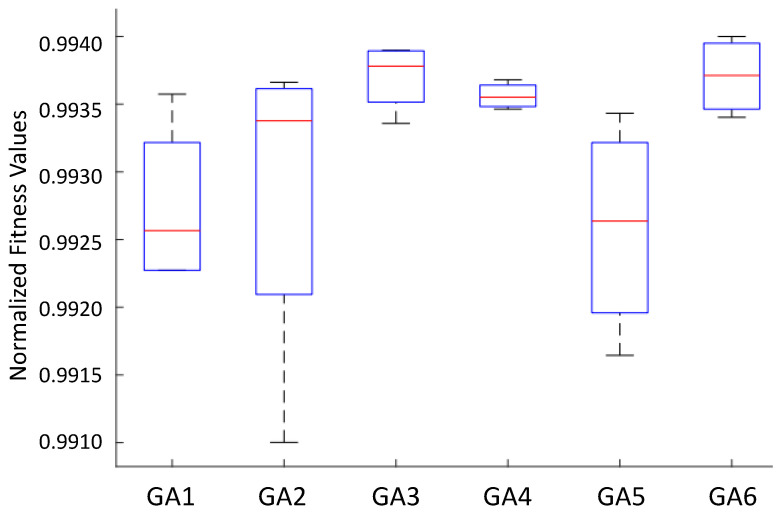
Boxplot of the six GA strategies.

**Figure 13 sensors-22-06094-f013:**
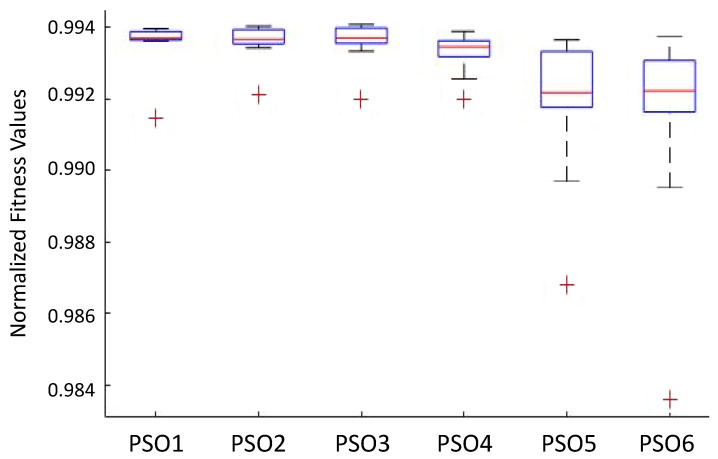
Boxplot of the six PSO strategies.

**Figure 14 sensors-22-06094-f014:**
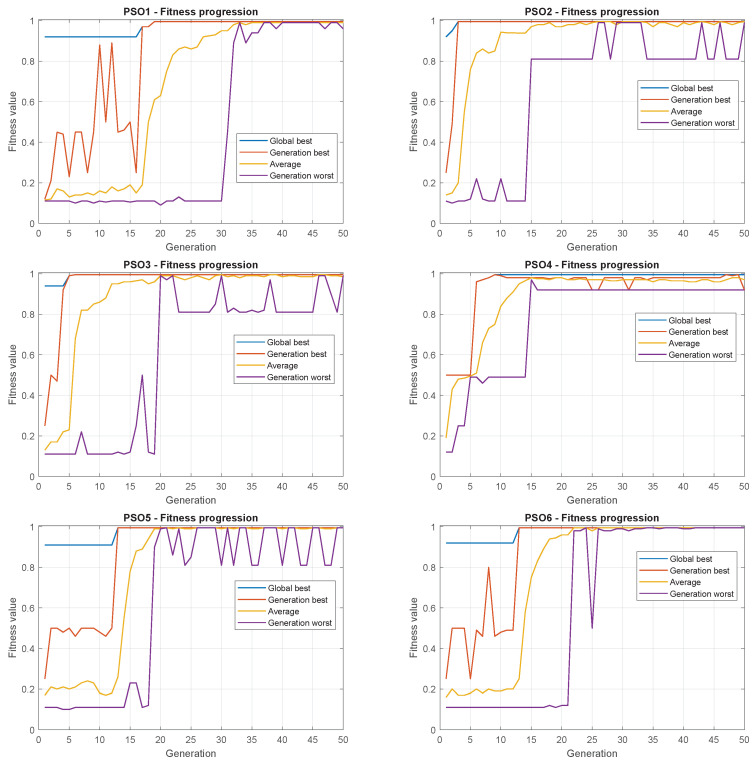
Evolution of PSO strategies.

**Figure 15 sensors-22-06094-f015:**
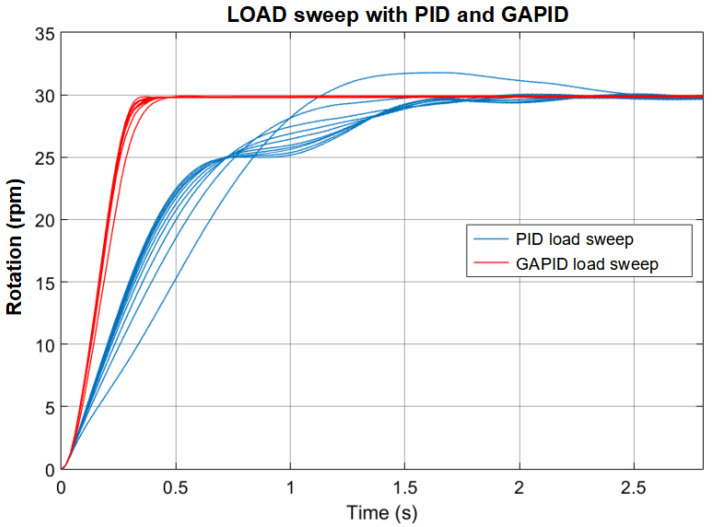
Transient responses of the system with PID and GAPID controllers subjected to variation of the moment of inertia *J* from 0.002 to 0.015.

**Figure 16 sensors-22-06094-f016:**
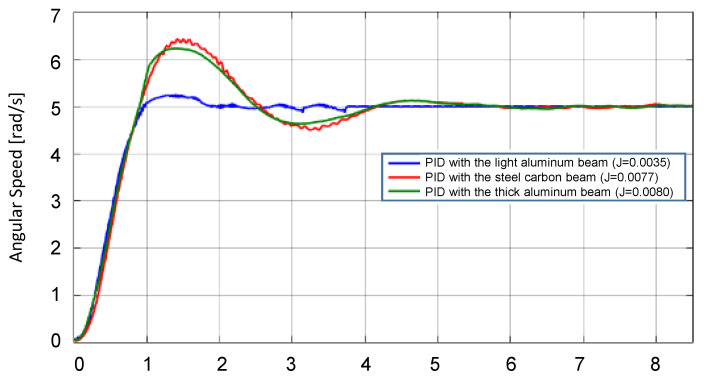
DC motor angular speed with PID control for different loads, to set point of 5 rad/s.

**Figure 17 sensors-22-06094-f017:**
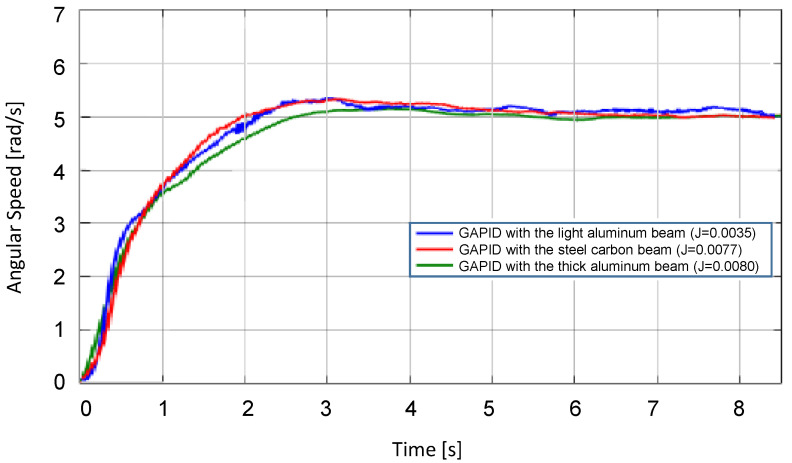
DC motor angular speed with GAPID control for different loads, to set point of 5 rad/s.

**Table 1 sensors-22-06094-t001:** Motor parameters.

Parameter	Value
$L_{a}$	0.0032 H
$R_{a}$	4.57 Ω
*B*	0.001405
*J*	depends on the beam
$k_{s}$	0.25
$k_{acd}$	0.02508
$k_{t}$	0.23309
$k_{e}$	0.23309

**Table 2 sensors-22-06094-t002:** Moments of inertia used in simulations.

Parameter	Value [kg·m^2^]
$J_{joint}$	0.00026
$J_{1}$	0.00250
$J_{2}$	0.00400
$J_{3}$	0.00550
$J_{4}$	0.00700
$J_{5}$	0.00850

**Table 3 sensors-22-06094-t003:** Moments of inertia for the three beams for experimental tests.

Beam (Load)	Inertia Value (J) [kg·m^2^]
Light slim aluminum	0.0035
Carbon steel beam	0.0077
Heavy thick aluminum beam	0.0080

**Table 4 sensors-22-06094-t004:** GA parameters.

Parameter	GA1	GA2	GA3	GA4	GA5	GA6
Sequence ^1^	S+C+M	S+C+M	S+C+M	S+C+M	C+M+S	C+M+S
Selection ^2^	RW	RW	BT	DT	RW	BT
Crossover probability	70%	100%	70%	70%	100%	100%
Mutation	5%					
Population	30					
Repetitions	10					

^1^ S = selection, C = crossover, M = mutation. ^2^ RW = roulette wheel, BT = binary tourneament, DT = death tourneament.

**Table 5 sensors-22-06094-t005:** The best fitness of each GA.

GA model	Fitness	Rank
GA1	0.993574	5
GA2	0.993661	4
GA3	0.993898	2
GA4	0.993680	3
GA5	0.993432	6
GA6	0.993999	1

**Table 6 sensors-22-06094-t006:** PSO parameters.

Parameter	PSO1	PSO2	PSO3	PSO4	PSO5	PSO6
Topology	Global			Ring		
ω	1	0.5	0.9→0.4	1	0.5	0.9→0.4
Search space	100					
Population	30					
Dimensions	6					
Maximum iterations	50					
$c_{1}$ and $c_{2}$	2.05					
Repetitions	10					

**Table 7 sensors-22-06094-t007:** The best fitness of each PSO.

PSO Model	Best Fitness	Rank
PSO1	0.992541	3
PSO2	0.992565	2
PSO3	0.992588	1
PSO4	0.992280	4
PSO5	0.990775	5
PSO6	0.990427	6

**Table 8 sensors-22-06094-t008:** Best GAPID parameters found (by PSO3).

Gain type	Gaussian Parameters		
Proportional	$k_{p0}=0.2$	$k_{p1}=100$	$q_{p}=0.9$
Integral	$k_{i0}=0.0637$	$k_{i1}=0.071$	$q_{i}=0.9$
Derivative		$k_{d1}=14$	$q_{d}=0.001$

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
