# Peer review of "Metaheuristics-Based Optimization of a Robust GAPID Adaptive Control Applied to a DC Motor-Driven Rotating Beam with Variable Load"

_sensors, 2022, doi:10.3390/s22166094_

Round 1
Reviewer 1 Report
[Comment 1] Novelty and literature review
[Subcomment 1a] (line 80) Please mention which previous works were meant.
[Subcomment 1b] If WOA is better in terms of the optimization result, why it is not considered in this study?
[Subcomment 1c] (lines 211-212) Please add literature review on studies that developed Simulink models for the same and similar problems with the one studied here.
[Subcomment 1d] (lines 232-234) It is necessary to state the reason why having the load variations as an additional dimension is important. Please state it at the novelty statement in the introduction section.
[Comment 2] Methods
[Subcomment 2a] (lines 119-120) It would be necessary to state what those nine-parameters are. The authors might want to relate the explanations with Equations (2)-(4).
[Subcomment 2b] Explanations of the genetic algorithm (GA) is unclear, e.g., (1) which chromosomes are dealt with when performing the crossover, then the mutation, (2) which set of chromosomes are used as candidates of parent chromosomes, etc. Please add a flowchart to explain the GA better. If necessary, please add another flowchart for the particle swarm optimization (PSO).
[Subcomment 2c] How is Equation (7) related to the gain function in Equations (1)-(4)?
[Subcomment 2d] (line 277) The authors need to briefly explain about the "ring topology".
[Comment 3] Experiments
[Subcomment 3a] (Section 4.1) Did the authors tune the parameters of GA and PSO? The results of the different algorithms are not comparable without appropriate parameter tuning. Even when using the same type of method, without the method's parameter tuning (for each of the 12 settings), the authors would not get the models to produce the best solutions. I saw Table 5 that only state one set of parameters, not 6 sets. Please revise all experiments if the authors have not conduct it yet. Please list the best parameters for each of the 12 models in a table, as a reference for next researchers.
[Subcomment 3b] (Figure 9) The authors need to explain how they calculate the normalized fitness values. How does it differ from Equation (8)?
[Subcomment 3c] When analyzing the results, the authors need to appropriately state their reasons, e.g., in lines 330-331, 332-333, 347-348, etc. The authors need to provide more intuitive explanations related to how the elements of the methods work.
[Subcomment 3d] It is common to run metaheuristics for several iterations to ensure the randomness and obtaining the average behavior of the algorithms. How many runs were conducted for each of GA models? If it was run only once, I suggest the authors run each model for 10 times, like what they did for PSO.
[Comment 4] References
The authors need to cite the reference where Equations (1)-(4) were taken from.
[Comment 5] Writing quality and clarity
[Subcomment 5a] (lines 34-35) Please place this sentence into the its previous paragraph. If possible, please avoid writing only one sentence in a paragraph. (Please check other sentences as well.)
[Subcomment 5b] Please define complete form of each abbreviation on its first use, e.g., WOA.
[Subcomment 5c] Please revise inappropriate use of capital letters, e.g., in line 60.
[Subcomment 5d] For clarity, please cite a reference for explaining the GA models (lines 248, 253, etc).
[Subcomment 5e] (Figure 8) The quality of the figure is very low. Please enlarge the figure until the font size is similar with the text and readable.
Author Response
Dear Associate Editor and Reviewers,
We are pleased to submit the Response to Reviewers of the manuscript Sensors-1799845 - “Metaheuristics-Based Optimization of a GAPID Adaptive Control Applied to a DC Motor-Driven Rotating Beam with Variable Load”.We appreciated the constructive criticisms of the reviewers.
We have addressed each of their concerns, as outlined below. We believe that the manuscript is substantially improved after making the suggested edits. We look forward to hearing from you regarding our submission, and we are open to respond to any further questions and comments you may have.
Sincerely yours,
Hugo Valadares Siqueira & Co-authors
======================================================================
[Comment 1] Novelty and literature review
[Subcomment 1a] (line 80) Please mention which previous works were meant.
Answer: Thanks for your suggestion: We included 3 references to these previous works.
======================================================================
[Subcomment 1b] If WOA is better in terms of the optimization result, why it is not considered in this study?
Answer: Good point. At first, this new investigation can be expanded to many other nature-inspired metaheuristics. In the work aforementioned we can note the WOA provided a slightly better result for the specific problem of GAPID controlling a DC-DC converter, but it took more iterations than PSO as well as a higher dispersion of the results. Our option was to use the most important evolutionary approach (GA) and the most prominent swarm-based method (PSO) because the literature is abundant in present variations of these methods. Note that we used 12 distinct methods to optimize the GAPID. Furthermore, several tests would be necessary for the inclusion of WOA or other metaheuristics. In this sense, we will leave it for future work. We included this comment in the conclusion section, around lines 447-451.
======================================================================
[Subcomment 1c] (lines 211-212) Please add literature review on studies that developed Simulink models for the same and similar problems with the one studied here.
Answer: There are other works which deal with modeling of rotating beams. In this sense, some references were added: [31-35], and referred in the text in lines 96 and 236.
======================================================================
[Subcomment 1d] (lines 232-234) It is necessary to state the reason why having the load variations as an additional dimension is important. Please state it at the novelty statement in the introduction section.
Answer: The statement "having the load variations as an additional dimension" can be misunderstood by the reader. So, we changed this part of the text by "adding an additional variable to the optimization problem" that is more clear.
In this sense, such statement was included in the introduction (around lines 78-82) and in the Methodology sections (around lines 232-234).
======================================================================
======================================================================
[Comment 2] Methods
[Subcomment 2a] (lines 119-120) It would be necessary to state what those nine-parameters are. The authors might want to relate the explanations with Equations (2)-(4).
Answer: It has been included “k_0, k_1 and \delta of eq. ([eq:fn-gaussiana]) for each of the three gains functions” in the text, around lines 126-129.
======================================================================
[Subcomment 2b] Explanations of the genetic algorithm (GA) is unclear, e.g., (1) which chromosomes are dealt with when performing the crossover, then the mutation, (2) which set of chromosomes are used as candidates of parent chromosomes, etc. Please add a flowchart to explain the GA better. If necessary, please add another flowchart for the particle swarm optimization (PSO).
Thank you for the valuable suggestion. We add new explanations in Sections 2.2.1 and 2.2.2, as well as the flowchart of the models. New descriptions in section 4.1.1. are added too.
======================================================================
[Subcomment 2c] How is Equation (7) related to the gain function in Equations (1)-(4)?
Answer: The parameterized second-order system is solely defined as a generic plant to test the damping factor sweep and the resulting fitness values of each fitness function (IAE, ISE and ITSE) for a step response in open loop. This test is important to demonstrate why IAE was chosen as fitness function in our work. The equations (1)-(4) does not refer to the plant but to the GAPID controller. In fact, they describe the gaussian functions of the adaptive gains of GAPID.
======================================================================
[Subcomment 2d] (line 277) The authors need to briefly explain about the "ring topology".
Answer: We agree with the reviewer, then the explanatory text was included:
“Two main topologies are addressed: a) Global Topology: each particle considers as global best (gbest) the best positioned particle for the sake of the entire population; b) Ring Topology: the particle considers as global best (gbest) the best positioned particle among its nearest topological neighbors, which are previously defined.” (around lines 219-222).
======================================================================
======================================================================
[Comment 3] Experiments
[Subcomment 3a] (Section 4.1) Did the authors tune the parameters of GA and PSO? The results of the different algorithms are not comparable without appropriate parameter tuning. Even when using the same type of method, without the method's parameter tuning (for each of the 12 settings), the authors would not get the models to produce the best solutions. I saw Table 5 that only state one set of parameters, not 6 sets. Please revise all experiments if the authors have not conduct it yet. Please list the best parameters for each of the 12 models in a table, as a reference for next researchers.
Answer: The authors thank the reviewer for the careful reading. The values used as parameters were selected considering a grid search, and we noted that small perturbations did not lead to important differences in the final results (sometimes in the 7th decimal place). In this specific topic we provided for each version of each algorithm the same conditions to achieve their global best. Note that we used as base fixed and well-known values because our main goal was to investigate the search capability considering already well-studied versions. For example, if we use distinct population sizes, we are dealing with unfair comparisons.
However, to allow the repetition of our experiments, we provided Tables 4 and 5 containing all the elements of the GA and PSO parameters. In this sense, we reformulated the table where the sets of parameters are more clear. Also, former Tables 4 and 6 were merged into the new Table 5, where the summary of all 12 models are gathered in a single table in order to simplify the visualization.
======================================================================
[Subcomment 3b] (Figure 9) The authors need to explain how they calculate the normalized fitness values. How does it differ from Equation (8)?
Answer: The normalized fitness is the one shown in eq. (8). It is referred as normalized because the top value approaches 1 as IAE tends to zero. This procedure is adopted by several authors to adjust a cost function (that must be minimized) into a fitness function (that must be maximized).
======================================================================
[Subcomment 3c] When analyzing the results, the authors need to appropriately state their reasons, e.g., in lines 330-331, 332-333, 347-348, etc. The authors need to provide more intuitive explanations related to how the elements of the methods work.
Answer: The authors appreciate the suggestion. As requested we discuss further the reasons why the algorithms behaved as we reported. The modifications are in lines 351 to 364.
======================================================================
[Subcomment 3d] It is common to run metaheuristics for several iterations to ensure the randomness and obtaining the average behavior of the algorithms. How many runs were conducted for each of GA models? If it was run only once, I suggest the authors run each model for 10 times, like what they did for PSO.
Answer: Well pointed. In fact, 10 simulations of each algorithm were performed and the average results were taken to compose the figures and tables shown in the paper. This is stated in the text only for PSO (line 344), but not for GAs. We included this information in the text around line 317. Furthermore, we also added Table 4 with a summary of the GAs following the model of table 5. Both tables also include the number of repetitions of each model.
======================================================================
[Comment 4] References
The authors need to cite the reference where Equations (1)-(4) were taken from.
Answer: We agree with the reviewer and included a reference to these equations. It is reference [1]:
Puchta, E.D.P.; Siqueira, H.V.; dos Santos Kaster, M. Optimization Tools Based on Metaheuristics 468 for Performance Enhancement in a Gaussian Adaptive PID Controller. IEEE Transactions on 469 Cybernetics 2019, pp. 1–10.
============================================================================================================================================
[Comment 5] Writing quality and clarity
[Subcomment 5a] (lines 34-35) Please place this sentence into the its previous paragraph. If possible, please avoid writing only one sentence in a paragraph. (Please check other sentences as well.)
Answer: Ok. This paragraph has been appended to the previous one. We revised all the paper and did the same for other sentences, avoiding short paragraphs, as possible.
======================================================================
[Subcomment 5b] Please define the complete form of each abbreviation on its first use, e.g., WOA.
Answer: Well pointed. We revised the whole paper and fixed this issue for all abbreviations.
======================================================================
[Subcomment 5c] Please revise inappropriate use of capital letters, e.g., in line 60.
Answer: We identified some misuses in the text and corrected them. But we prefered to keep some words with capital letters because they define concepts, techniques, algorithms etc.
======================================================================
[Subcomment 5d] For clarity, please cite a reference for explaining the GA models (lines 248, 253, etc).
Answer: We fully agree with the reviewer. We add new references related to the GA together with the flowcharts requested in Subcomment 2b.
======================================================================
[Subcomment 5e] (Figure 8) The quality of the figure is very low. Please enlarge the figure until the font size is similar with the text and readable
Answer: We agree with the reviewer. The figures of the evolution of GA and PSO were rebuilt and converted previously to PDF to enable high resolution when zoomed. The size of the letters were a bit enlarged and are more readable than before.

Reviewer 2 Report
1. In the introduction, state the research question clearly and list the main contributions of the paper.
2. Avoid using we/our throughout the paper.
3. The paper is dealing with swarm intelligence metaheuristics optimization, yet only two algorithms and one application (DC motor) are mentioned.
Mention several other representative algorithms, and also other applications, and include the following:
https://www.mdpi.com/1424-8220/22/5/1711
https://link.springer.com/article/10.1007/s00521-019-04441-0
https://onlinelibrary.wiley.com/doi/abs/10.1002/ett.3770
4. Mention that the problem of parameters tuning is belonging to the NP-hard challenges, that cannot be addressed with traditional deterministic methods, before introducing the stochastics metaheuristics approach.
5. Elaborate why the GA and PSO were selected and justify it. There are numerous recent, powerful swarm intelligence metaheuristics available.
6. Consider adding more algorithms to the comparative analysis (FA, ABC, BAT, etc).
7. Discuss the limitations of the proposed approach.
Author Response
Dear Associate Editor and Reviewers,
We are pleased to submit the Response to Reviewers of the manuscript Sensors-1799845 - “Metaheuristics-Based Optimization of a GAPID Adaptive Control Applied to a DC Motor-Driven Rotating Beam with Variable Load”.We appreciated the constructive criticisms of the reviewers.
We have addressed each of their concerns, as outlined below. We believe that the manuscript is substantially improved after making the suggested edits. We look forward to hearing from you regarding our submission, and we are open to respond to any further questions and comments you may have.
Sincerely yours,
Hugo Valadares Siqueira & Co-authors
======================================================================
- In the introduction, state the research question clearly and list the main contributions of the paper.
Answer: Some modifications in the introduction section (around lines 76-91) were made in order to state the contribution of this work more clearly.
======================================================================
- Avoid using we/our throughout the paper.
Answer: Done: all sentences in the text with we/our have been changed.
======================================================================
- The paper is dealing with swarm intelligence metaheuristics optimization, yet only two algorithms and one application (DC motor) are mentioned.
Mention several other representative algorithms, and also other applications, and include the following:
https://www.mdpi.com/1424-8220/22/5/1711
https://link.springer.com/article/10.1007/s00521-019-04441-0
https://onlinelibrary.wiley.com/doi/abs/10.1002/ett.3770
Answer: Good suggestion. As requested, we mention those and other new works in our manuscript and we included the comment about the importance of such investigations. Thank you.
======================================================================
- Mention that the problem of parameters tuning is belonging to the NP-hard challenges, that cannot be addressed with traditional deterministic methods, before introducing the stochastics metaheuristics approach.
Answer: The recommended text has been included in the introduction section, around line 62.
======================================================================
- Elaborate why the GA and PSO were selected and justify it. There are numerous recent, powerful swarm intelligence metaheuristics available.
Answer: Good point. Our option was to use the most important evolutionary approach (GA) and the most prominent swarm-based method (PSO) because the literature is abundant in variations of the aforementioned methods. Note that we used 12 single variations to optimize the GAPID. Furthermore, several tests would be necessary for the inclusion of other metaheuristics. In this sense, we will leave it for future work. We included this comment in the conclusion section, around lines 447-451.
======================================================================
- Consider adding more algorithms to the comparative analysis (FA, ABC, BAT, etc).
As discussed in the previous question, we included the use of other optimizers as future works.
======================================================================
- Discuss the limitations of the proposed approach.
Answer: Good point. A piece of text around lines 93-100 that explains the limitations of the proposed technique, which are the known limitations of PID control, and by the impractical use of optimization algorithms directly in a real-world plant, demanding the use of computational models for the tests to obtain all the fitness values and perform the optimization.

Round 2
Reviewer 1 Report
Thank you for your revisions.
Reviewer 2 Report
The authors have addressed all the issues from the previous round, and modified the paper accordingly. It can be accepted in present form.